# Costs of implementing universal test and treat in three correctional facilities in South Africa and Zambia

Rachel Mukora[1]*, Helene J. Smith[2,3], Michael E. Herce[2,4], Lucy Chimoyi[1], Harry Hausler[5], Katherine L. Fielding[6,7], Salome Charalambous[1,7], Christopher J. Hoffmann[1,8]

1 The Aurum Institute, Aurum House, The Ridge, Johannesburg, South Africa, 2 Centre for Infectious Disease Research in Zambia (CIDRZ), Lusaka, Zambia, 3 School of Public Health & Community Medicine, University of New South Wales, Sydney, Australia, 4 Institute for Global Health and Infectious Diseases, University of North Carolina at Chapel Hill, Chapel Hill, NC, United States of America, 5 TB/HIV Care Association, Cape Town, South Africa, 6 London School of Hygiene & Tropical Medicine, London, United Kingdom, 7 School of Public Health, University of the Witwatersrand, Johannesburg, South Africa, 8 Department of Epidemiology, Johns Hopkins Bloomberg School of Public Health, Baltimore, MD, United States of America

* rmukora@auruminstitute.org

**Data Availability Statement:** All relevant data are within the paper and its Supporting Information files.

## Abstract

### Introduction

Universal test and treat (UTT) is a population-based strategy that aims to ensure widespread HIV testing and rapid antiretroviral therapy (ART) for all who have tested positive regardless of CD4 count to decrease HIV incidence and improve health outcomes. Little is known about the specific resources required to implement UTT in correctional facilities for incarcerated people. The primary aim of this study was to describe the resources used to implement UTT and to provide detailed costing to inform UTT scale-up in similar settings.

### Methods

The costing study was a cross-sectional descriptive study conducted in three correctional complexes, Johannesburg Correctional Facility in Johannesburg (>4000 inmates) South Africa, and Brandvlei (~3000 inmates), South Africa and Lusaka Central (~1400 inmates), Zambia. Costing was determined through a survey conducted between September and December 2017 that identified materials and labour used for three separate components of UTT: HIV testing services (HTS), ART initiation, and ART maintenance. Our study participants were staff working in the correctional facilities involved in any activity related to UTT implementation. Unit costs were reported as cost per client served while total costs were reported for all clients seen over a 12-month period.

### Results

The cost of HIV testing services (HTS) per client was $ 92.12 at Brandvlei, $ 73.82 at Johannesburg, and $ 65.15 at Lusaka. The largest cost driver for HIV testing at Brandvlei were

**Funding:** This work was supported by funding from the U.K. Department for International Development (DFID)/ UKAID under grant MMM/EHPSA/AURUM/05150013. The contents of this manuscript are the sole responsibility of the study authors and do not necessarily reflect the views of DFID, UKAID, or the United Kingdom government. The funding agencies had no role in the study design, data collection or analysis, manuscript writing, or the decision to submit for publication.

**Competing interests:** The authors have declared that no competing interests exist.

staff costs at 55.6% of the total cost, while at Johannesburg (56.5%) and Lusaka (86.6%) supplies were the largest contributor. The cost per client initiated on ART was $917 for Brandvlei, $421.8 for Johannesburg, and $252.1 for Lusaka. The activity cost drivers were adherence counselling at Brandvlei (59%), and at Johannesburg and Lusaka it was the actual ART initiation at 75.6% and 75.8%, respectively. The annual unit cost for ART maintenance was $2,640.6 for Brandvlei, $710 for Johannesburg, and $385.5 for Lusaka. The activity cost drivers for all three facilities were side effect monitoring, and initiation of isoniazid preventive treatment (IPT), cotrimoxazole, and fluconazole, with this comprising 44.7% of the total cost at Brandvlei, 88.9% at Johannesburg, and 50.5% at Lusaka.

## Conclusion

Given the needs of this population, the opportunity to reach inmates at high risk for HIV, and overall national and global 95-95-95 goals, the UTT policies for incarcerated individuals are of vital importance. Our findings provide comparator costing data and highlight key drivers of UTT cost by facility.

## Introduction

Universal test and treat (UTT) for HIV is a population-based strategy that aims to ensure widespread HIV testing and rapid antiretroviral therapy (ART) for all who have tested HIV-positive regardless of CD4 count to decrease HIV incidence and improve health outcomes. UTT has been tested in several settings and adopted as part of national HIV programmes in many countries, including both South Africa and Zambia. UTT was introduced in South Africa in September 2016 and recommended same-day ART initiation from October 2017 [1, 2]. Prior to UTT, one retrospective study done using programmatic data in South Africa described viral load suppression at six and 12 months after ART initiation at 94.7% and 92.5% respectively for incarcerated people with HIV treated in an on-site ART programme [3]. A cross sectional study done in Malawi also prior to UTT, found 95% viral suppression among a sample of incarcerated people with HIV who received ART for at least 6 months in a prison clinic [4]. The TasP study conducted in South Africa and Zambia showed that it is feasible to implement UTT in diverse southern African correctional settings [5]. The study observed 86% ART uptake among incarcerated participants, many of whom initiated ART on the same day as HIV testing, and retention in care and viral load suppression exceeding 90% at 6 months post-ART initiation for individuals who remained incarcerated [5].

Current studies examining cost for HIV care typically include general populations; little is known about the specific resources required to implement UTT in correctional facilities [6–8]. A modelling study based on general population data from South Africa (SA) in 2014 reported that universal treatment averted more HIV infections, but had a higher total cost of about $320 per quality adjusted life year (QALY) gained compared to the status quo (providing ART based on an initiation threshold of CD4 count $\leq$350 cells/mm$^3$), which had a total cost of around $290 per QALY gained [9]. Another modelling study (2014) based on data from SA, Zambia, Vietnam and India reported that UTT met cost-effectiveness thresholds [10]. In SA, the cost per Disability Adjusted Life Year (DALY) averted for changing eligibility to all HIV-positive adults compared with eligibility for those with CD4 counts of $\leq$350 cells per μL ranged

from $438 to $3790 in seven models used while in Zambia results showed $790 per DALY in four mathematical models [10].

The primary aim of this study was to describe the resources used to implement UTT in correctional settings as part of the Treatment as Prevention (TasP) study [5]. TasP was an implementation research study designed to assess the feasibility of UTT delivery in three correctional complexes, two in SA and one in Zambia. Through this study, we aimed to provide costing to inform UTT scale-up in similar settings.

## Methods

Details on how inmates at each level of the HIV cascade engaged with HIV self-services have been detailed elsewhere [5]. The TasP study provided voluntary universal HCT at facility entry, and through HCT campaigns to reach all inmates at least once annually and offered ART to all inmates who tested HIV-positive. The TasP study encouraged increased HIV services uptake and assured universal HCT and ART access. The TasP study augmented ART care, offering ART initiation to all inmates (after screening for TB and kidney disease) regardless of CD4 count.

### Study design and settings

TasP study findings on clinical and select implementation outcomes have been reported previously [5, 11]. The costing component of the TasP study was a cross-sectional descriptive study to describe the resources needed to implement UTT in two correctional facilities in South Africa: Johannesburg Correctional Facility in Johannesburg (>4000 inmates), and Brandvlei Correctional Facility in Western Cape (~3000 inmates) and Lusaka Central in Lusaka, Zambia (~1400 inmates). The three correctional complexes collectively encompassed ten correctional units, which included six male units (two maximum security), three female units, and one youth unit (for people aged 18–22 years) [5]. TasP sites were selected purposively to reflect a range of security levels and sociodemographic characteristics of incarcerated people in southern Africa [5]. Each site had routine health services available on site, including HIV testing and treatment [5].

In all three facilities there were specific structured management systems in place prior to initiation of the UTT. In Brandvlei and Johannesburg, unit managers managed doctors, nurses, and peer educators providing services in their clinical area. Each unit manager reported to a HIV/AIDS co-ordinator who reported to the health service manager. All health service managers reported to a central area co-ordinator who had responsibility for a management area that included several correctional facilities. The rationale for the staffing in SA was based on the national policy regarding staff per facility rather than on need.

Lusaka had fewer managerial layers but had a clinic manager providing line management to doctors, nurses, and peer educators (Table 1).

### Study participants

We included staff working in the correctional facilities involved in any activity related to UTT implementation. Staff cadres included managers, HIV co-ordinators, counsellors, adherence counsellors, medical doctors, professional nurses, peer educators and security officials including those employed by the correctional services, the health departments, and non-governmental organizations (NGOs). We purposively selected all staff within the cadre where there were two or less staff in the role. In cadres where three or more staff performed the same duties, we used convenience sampling where every 2nd person was selected.

**Table 1. Staff involved in HTS at correctional facilities.**

| Brandvlei Correctional | | Johannesburg Correctional | | Lusaka Central | |
|---|---|---|---|---|---|
| Cadre | Number | Cadre | Number | Cadre | Number |
| Operations manager | 5 | Health manager | 1 | Offender manager | 1 |
| Head of Centre | 1 | Clinic in charge | 2 | Clinic in charge | 1 |
| HIV coordinator | 3 | HIV coordinator | 1 | HIV coordinator | 1 |
| Medical doctor | 1 | Medical doctor | 2[a] | Clinical officer | 4 |
| Professional nurse | 7 | Professional nurse | 18 | Professional nurse | 1 |
| | | | | Enrolled nurse | 6 |
| Professional nurse counsellor | 1 | Adherence counsellor | 2 | Adherence counsellor | 2 |
| | | Pharmacist | 5 | Pharmacist | 2 |
| HTS counsellor/Health screener | 4 | HTS counsellor | 10 | HTS counsellor/Health screener | 5 |
| Officer in charge | 1 | | | Officer in charge | 1 |
| Case officer | 4 | | | | |
| | | Peer educator | 10 | Peer educator | 10 |
| Security officer | 10 | Security officer | 7 | Duty officer | 5 |
| Warden | 2 | | | Cell captain | 32 |
| | | Project manager | 1 | Project manager | 1 |
| Project coordinator | 1 | Project coordinator | 1 | Project coordinator | 1 |
| Data capturers | 2 | Data manager | 1 | Data associate | 2 |
| | | Data monitor | 1 | Xray/Xpert technician | 3 |
| **Total staff** | **42** | | **62** | | **78** |
| **Total inmates** | **3,000** | | **4,000** | | **1,400** |
| **Staff: Inmates ratio** | **1:71** | | **1:65** | | **1:18** |
| **Total correctional units** | **2** | **Total correctional units** | **3** | **Total correctional units** | **3** |
| **Male and Female units co-joined (medium security)** | **2** | **Male (maximum security)** | **1** | **Male (maximum security)** | **1** |
| | | **Male (medium security)** | **1** | **Male (medium security)** | **1** |
| | | **Female units** | **1** | Youth units | **1** |
| **Estimated HIV Prevalence** | **5%** | | **15%** | | **15%** |
| **Clients initiated on ART** | **35** | | **233** | | **229** |

[a] First medical doctor at 100% FTE (Full Time Equivalent) and second medical doctor at 60% FTE

## Costing methods

**Data collection.** Costing was determined through a survey conducted between September and December 2017 that we used to identify materials and labour used for three separate components of UTT: HIV testing services (HTS), ART initiation, and ART maintenance. This approach considered resource-use in terms of cost of each item: salaries, equipment, laboratory tests and medications. The other consideration was the overall infrastructure present at the time of program implementation. National and facility health management were in place in both South Africa and Zambia, allowing the UTT program to be placed within an existing health infrastructure. In addition, logistics for supplies, off-site laboratory testing, and testing for incarcerated people were in place. This program augmented staff and supplies for service delivery without a need to develop a management structure, clinical infrastructure, or logistics capabilities.

A standardized interview tool developed by investigators was used to assess four main inputs across the three facilities: staff, equipment, supplies and medical tests. The paper-based tool was administered by trained research assistants in a private setting within the correctional facility and the interview lasted between 30–60 minutes.

Costs were determined through record reviews of invoices and price lists for both South Africa and Zambia. We used a combination of top-down and bottom-up micro-costing approaches to estimate the resources/inputs that were used for UTT. The inputs were costed over a 12-month period. The outcomes of interest were unit and total costs per activity for each facility and were presented from a provider perspective.

**Data analysis.** For all cadres interviewed, the main activity categories included HIV testing, ART (antiretroviral) initiation and ART maintenance. HIV testing included the following sub-activities, namely: group counselling, pre-test counselling, testing/screening for HIV and counselling after result. ART initiation included: adherence counselling, phlebotomy and initial ART prescription. ART maintenance involved: phlebotomy, educational messages on HIV treatment and adherence and ART maintenance (assessing for side effects of ART, isoniazid preventive therapy, cotrimoxazole, and fluconazole and considering preventive therapies). Allocation of costs for phlebotomy between ART initiation and ART maintenance was based on the proportions of clients seen for each visit type. All time spent on research activities was excluded from the analysis.

Data were entered into REDCap software version 7.6.9 (Vanderbilt University) and analysis was completed using MS Excel (Microsoft Corp, USA, 2003). We calculated unit costs as cost per client served. All costs were converted into the US dollar (USD) using exchange rates on 1 January 2017: 1USD = 13.78 South African Rand (ZAR) for the South African sites and 1USD = 9.90 Zambian Kwacha (ZMW) for the Zambian site. Costing results are reported separately for each facility and as overall cost for care provision and as cost per client served (unit cost). Notably, these costs are not related to a specific payer as the budget for various HIV testing, HIV care, and program costs came from a variety of sources including the grant supporting this project, national health and/or corrections budgets, and donor funding.

*Staff*. Study team members administered a costing instrument to staff to describe activities and time spent over the preceding work week (Monday to Friday). The survey included questions regarding the amount of the workday spent on an activity, the time spent per client to complete the activity (estimated and self-reported), and all equipment and supplies used during the activity.

To calculate costs over a 12-month period, we used the proportion of time spent on implementation activities and the annual salary (base salary plus fringe benefits) for NGO staff and the median annual salary for corrections staff. Salaries in SA were obtained from the Department of Public Service and Administration and the NGO Finance department for staff employed by the NGO. Salaries in Zambia were obtained from the correctional facility. Staff unit costs were computed by dividing the annual total costs with the number of clients seen over a 12-month period which we obtained from the TasP study (S1 Table) for a period of 17 months, though we based our costs on the average number of clients seen over a 12-month period. For ART maintenance, we excluded incarcerated persons who were released, transferred or died during the study period and costed only those who remained incarcerated and on ART.

*Equipment*. All staff involved in UTT were asked to state the quantities of all the equipment they used and the activities they performed and to specify whether the equipment was shared and with whom and for what other tasks. Costs were then based on the proportion of time the equipment was used for UTT and the market price or invoices. The resulting costs for the equipment were annuitized using a discount rate of 10.5% for both South Africa and Zambia [12]. The same useful life was used in both countries, with 10 years for examination tables, filing cabinets and weighing scales, and 5 years for computers, chairs, desks and blood pressure machines. To avoid double counting the costs of any shared equipment, cost per minute estimates were derived using the time spent data and applied to the annuitized cost of the

equipment. We considered this approach bottom-up since cost per minute estimates were used to calculate the annual cost of the equipment used [13].

*Supplies*. Costs for supplies used for UTT implementation were determined using the most complete package of items mentioned by the participants as needed for the activity. This was a top-down approach that relied on interviews and item cost abstraction and did not use direct observation. Supplies were categorised as: HIV testing, ART initiation, and ART maintenance. The cost of antiretroviral therapy drugs over the 12-month period was also included. For ART initiation, a one month's supply of ART was costed since the incarcerated individuals returned after one month for their next supply. For ART maintenance, we costed over the remaining 11 months. ART maintenance costs did not include any additional acute or chronic care medical services clients received, but included HIV lab monitoring costs such as viral load testing.

*Medical tests*. A standard package of medical tests was included in the costing for both South Africa and Zambia. During ART initiation, clients receive three baseline tests: CD4, Creatinine and rapid plasma reagin (RPR) test. During ART maintenance, all clients on ART are meant to receive the following tests: creatinine (3, 6 & 12 months), HIV RNA Viral Load (6 & 12 months), while an estimated 5% (based on empiric data from these facilities) of clients receive haemoglobin, alanine transferase (ALT) and Xpert MTB/RIF testing for specific HIV care related indications. Unit prices for each test were obtained from local laboratories.

## Ethics approval

This study was approved by the Human Research Ethics Committees/Institutional Review Boards of: University of Witwatersrand (South Africa); the University of the Western Cape (South Africa); the University of Zambia (Zambia); the University of North Carolina at Chapel Hill, the University of Alabama at Birmingham, and Johns Hopkins University all in the USA; the London School of Hygiene and Tropical Medicine (UK); and James Cook University (Australia). The study was also approved by the Department of Correctional Services in South Africa and the Zambia Correctional Service. Written informed consent was obtained from all study participants in the study approved languages namely English or Afrikaans for South Africa and English, Nyanja and Bemba for Zambia. No reimbursements or incentives were offered to participants.

## Results

For each staff cadre we interviewed 1–5 individuals at each site. A total of 107 staff were interviewed with the following breakdown: in Johannesburg, 37 interviews (enrolment target 37); in Brandvlei, 31 interviews were conducted (target 31); and in Lusaka, 39 interviews were conducted (target 40).

## Costing

The cost of HIV testing services (HTS) for one client was $ 92.12 at Brandvlei, $ 73.82 at Johannesburg, and $ 65.15 at Lusaka (Table 2). The cost per client initiated on ART was $917 for Brandvlei, $421.8 for Johannesburg, and $252.1 for Lusaka (Table 3). The annual unit cost for ART maintenance was $2,640.6 for Brandvlei, $710 for Johannesburg, and $385.5 for Lusaka (Table 4). Fig 1 shows a summary of the cost per client for HTS, ART initiation and ART maintenance. In addition, the unit cost of gathering incarcerated people to be tested and escorting them for each clinic visit costs $6.56 in Brandvlei, $1.18 in Johannesburg and $0.77 in Lusaka.

For ART initiation, the main input cost drivers were staff costs at both Brandvlei (86%) and Johannesburg (72%), while baseline medical laboratory testing were the largest at Lusaka

**Table 2. Unit[a] and total costs[b] ($US) of HIV testing services (HTS) at correctional facilities in 2017.**

| | | Brandvlei Correctional | | | | Johannesburg Correctional | | | | Lusaka Central | | | |
|---|---|---|---|---|---|---|---|---|---|---|---|---|---|
| | | Staff | Equipment[c] | Supplies[d] | Total | Staff | Equipment[c] | Supplies[d] | Total | Staff | Equipment[c] | Supplies[d] | Total |
| **Group counselling** | Unit cost | 11.41 | 0.03 | 0.37 | **11.81** | 2.78 | 0.13 | 0.82 | **3.73** | 2.55 | 0.09 | 0.52 | **3.16** |
| | | (22.3%) | (8.1%) | (0.9%) | **(12.8%)** | (8.7%) | (50.0%) | (2.0%) | **(5.1%)** | (32.2%) | (10.8%) | (0.9%) | **(4.9%)** |
| | Total cost | 21 685 | 48 | 712 | **22 445** | 8 507 | 385 | 2 496 | **11 388** | 6 325 | 230 | 1 279 | **7 834** |
| **Pre-test counselling** | Unit cost | 8.47 | 0.07 | 0.04 | **8.58** | 8.31 | 0.04 | 0.42 | **8.77** | 1.58 | 0.21 | 7.08 | **8.87** |
| | | (16.5%) | (18.9%) | (0.1%) | **(9.3%)** | (26.1%) | (15.4%) | (1.0%) | **(11.9%)** | (20.0%) | (25.3%) | (12.6%) | **(13.6%)** |
| | Total cost | 16 110 | 124 | 85 | **16 319** | 25 409 | 130 | 1 283 | **26 822** | 3 928 | 523 | 17 699 | **22 150** |
| **Testing for HIV** | Unit cost | 25.95 | 0.26 | 40.07 | **66.28** | 12.46 | 0.04 | 40.19 | **52.69** | 2.12 | 0.37 | 48.79 | **51.28** |
| | | (50.7%) | (70.3%) | (98.9%) | **(71.9%)** | (39.1%) | (15.4%) | (96.4%) | **(71.4%)** | (26.8%) | (44.6%) | (86.5%) | **(78.7%)** |
| | Total cost | 49 338 | 492 | 27 627 | **77 458** | 38 113 | 128 | 54 765 | **93 006** | 5 263 | 914 | 114 066 | **120 244** |
| **Counselling after result** | Unit cost | 5.40 | 0.01 | 0.04 | **5.45** | 8.31 | 0.05 | 0.27 | **8.63** | 1.66 | 0.16 | 0.02 | **1.84** |
| | | (10.5%) | (2.7%) | (0.1%) | **(5.9%)** | (26.1%) | (19.2%) | (0.6%) | **(11.7%)** | (21.0%) | (19.3%) | (0.0%) | **(2.8%)** |
| | Total cost | 10 264 | 23 | 69 | **10 355(** | 25 409 | 143 | 813 | **26 364** | 4 119 | 389 | 48 | **4 557** |
| **Total** | **Unit cost** | **51.23** | **0.37** | **40.52** | **92.12** | **31.86** | **0.26** | **41.70** | **73.82** | **7.91** | **0.83** | **56.41** | **65.15** |
| | | **(55.6%)** | **(0.4%)** | **(44.0%)** | **(100%)** | **(43.2%)** | **(0.4%)** | **(56.5%)** | **(100%)** | **(12.1%)** | **(1.3%)** | **(86.6%)** | **(100%)** |
| | **Total cost** | **97 397** | **686** | **28 493** | **126 576** | **97 437** | **786** | **59 357** | **157 580** | **19 635** | **2 057** | **133 092** | **154 784** |

[c]Equipment and
[d]supplies lists are detailed in supporting information S2 and S3 Tables

(58.7%). The activity cost driver was adherence counselling at Brandvlei (59%), whereas at Johannesburg and Lusaka it was the initial ART prescription at 75.6% and 75.8% respectively.

For ART maintenance, the main input cost drivers were staff costs at Brandvlei (88.9%) and Johannesburg (53.6%) and supplies at Lusaka (64.8%). The activity cost drivers at all three facilities were side effect monitoring and initiation of Isoniazid Preventive Therapy (IPT), cotrimoxazole, and fluconazole, with this comprising 44.7% of the total cost at Brandvlei, 88.9% at Johannesburg, and 50.5% at Lusaka.

## Discussion

In this descriptive costing study, we provide cost estimates for delivering HTS, ART initiation, and ART maintenance for UTT implementation in three diverse correctional settings in Zambia and South Africa. Notably, we observed substantial variation in cost across these three correctional complexes which was largely related to the ratio of health staff to the inmate population, and, to a lesser extent, to variation in staff salary between South Africa and Zambia. We believe that these findings can help guide considerations for resource allocation for implementing UTT in similar correctional settings.

The costs that we estimated are higher than some prior reports for HIV care in the community [6, 7]. For example, HTS during PMTCT in South Africa has been estimated to cost between $31 and $38 and, in Zambia, $19 per client tested [6, 7]. This is in contrast to the unit costs we found for HTS in correctional settings, which ranged from $65.15 in Zambia to $73.82 in Johannesburg and $92.12 in Brandvlei in South Africa. There are several reasons why these costs are likely higher than community health facility HTS costs. First, additional resources are required for security within a correctional facility environment due to the

**Table 3. Unit[a] and total costs[b] ($US) of ART initiation at correctional facilities in 2017.**

| | Brandvlei Correctional | | | | | Johannesburg Correctional | | | | | Lusaka Central | | | | |
|---|---|---|---|---|---|---|---|---|---|---|---|---|---|---|---|
| | Staff | Equipment[c] | Supplies[d] | Medical Tests | Total | Staff | Equipment[c] | Supplies[d] | Medical Tests | Total | Staff | Equipment[c] | Supplies[d] | Medical Tests | Total |
| **Adherence counselling** | 541.9 | 2.4 | 0.1 | | 544.4 | 62.8 | 0.6 | 0.5 | | 63.9 | 11.4 | 0.4 | 0.0 | | 11.8 |
| | (68.7%) | (20.7%) | (0.2%) | | (59.4%) | (20.7%) | (58.3%) | (2.0%) | | (15.1%) | (20.5%) | (30.2%) | (0.00%) | | (4.7%) |
| | 22 951 | 103 | 3 | | 23 058 | 28 166 | 270 | 204 | | 28 640 | 4 965 | 169 | 0 | | 5 134 |
| **Phlebotomy** | 222.1 | 2.0 | 3.4 | | 227.4 | 39.1 | 0.4 | 3.6 | | 43.1 | 10.0 | 0.4 | 38.7 | | 49.1 |
| | (28.2%) | (16.8%) | (58.7%) | | (24.8%) | (12.9%) | (35.9%) | (15.8%) | | (10.2%) | (18.0%) | (28.7%) | (81.9%) | | (19.5%) |
| | 9 405 | 83 | 1 138 | | 10 626 | 17 541 | 164 | 1 617 | | 19 322 | 2 284 | 162 | 16 842 | | 19 288 |
| **ART initiation** | 24.8 | 7.4 | 18.8 | 94.2 | 145.2 | 201.9 | 0.1 | 18.7 | 94.2 | 314.8 | 34.2 | 0.5 | 8.6 | 147.9 | 191.2 |
| | (3.1%) | (62.5%) | (41.1%) | (100%) | (15.8%) | (66.4%) | (5.8%) | (82.1%) | (100%) | (74.6%) | (61.5%) | (41.1%) | (18.1%) | (100%) | (75.8%) |
| | 877 | 259 | 665 | 1 031 | 2 832 | 47 027 | 15 | 4 358 | 6 805 | 58 204 | 7 808 | 121 | 1 956 | 9 561 | 19 446 |
| **Total** | 788.8 | 11.8 | 22.3 | 94.2 | 917.0 | 303.9 | 1.0 | 22.8 | 94.2 | 421.8 | 55.6 | 1.3 | 47.3 | 147.9 | 252.1 |
| | (86.0%) | (1.5%) | (5.8%) | (11.8%) | (100%) | (72.0%) | (0.2%) | (5.4%) | (22.3%) | (100%) | (22.0%) | (0.5%) | (18.8%) | (58.7%) | (100%) |
| | 33 232 | 446 | 1 807 | 1 031 | 36 516 | 92 734 | 449 | 6 179 | 6 805 | 106 167 | 15 056 | 452 | 18 798 | 9 561 | 43 868 |

[a]Unit costs are showed on the first row of each activity while

[b]total costs are shown on the second row

[c]Equipment and

[d]supplies lists are detailed in supporting information S2 and S3 Tables

Table 4. Unit[a] and total costs[b] ($US) of ART maintenance at correctional facilities in 2017.

| | Brandvlei Correctional | | | | | Johannesburg Correctional | | | | | Lusaka Central | | | | |
|---|---|---|---|---|---|---|---|---|---|---|---|---|---|---|---|
| | Staff | Equipment[c] | Supplies[d] | Medical Tests | Total | Staff | Equipment[c] | Supplies[d] | Medical Tests | Total | Staff | Equipment[c] | Supplies[d] | Medical Tests | Total |
| Phlebotomy | 137.7 | 1.2 | 13.4 | | 152.3 | 24.2 | 0.6 | 14.4 | | 39.2 | 3.6 | 0.1 | 155.0 | | 158.7 |
| | (5.9%) 5 830.8 | (9.9%) 51.7 | (5.5%) 294.1 | | (5.8%) 6 176.6 | (6.4%) 10 843.7 | (7.6%) 101.3 | (5.1%) 2 078.1 | | (5.5%) 13 023.0 | (6.1%) 817.5 | (4.0%) 58.0 | (62.1%) 12 679.3 | | (41.2%) 13 554.8 |
| Educational messages | 1 301.1 | 4.8 | 2.7 | | 1 308.6 | 36.7 | 1.9 | 0.8 | | 39.3 | 30.9 | 0.8 | 0.5 | | 32.2 |
| | (55.4%) 45 921.2 | (39.1%) 221.3 | (1.1%) 59.0 | | (49.6%) 46 201.4 | (9.6%) 8 537.7.1 | (24.0%) 432.2 | (0.3%) 117.5 | | (5.5%) 9 087.4 | (52.4%) 7 072.9 | (23.5%) 175.9 | (0.2%) 42.2 | | (8.4%) 7 291.0 |
| ART maintenance | 908.1 | 6.3 | 228.3 | 36.9 | 1 179.6 | 319.8 | 5.3 | 269.4 | 36.9 | 631.4 | 24.6 | 2.4 | 94.1 | 73.5 | 194.6 |
| | (38.7%) 19 871.6 | (51.0%) 425.3 | (93.4%) 4 996.4 | (100%) 2 038.8 | (44.7%) 27 332.0 | (84.0%) 46 056.6 | (68.4%) 762.6 | (94.6%) 38 788.6 | (100%) 13 455.8 | (88.9%) 99 063.6 | (41.6%) 2 011.6 | (72.6%) 194.6 | (37.7%) 7 700.8 | (100%) 21 223.0 | (50.5%) 31 130.0 |
| Total | 2 346.9 | 12.3 | 244.5 | 36.9 | 2 640.6 | 380.7 | 7.8 | 284.6 | 36.9 | 710.0 | 59.1 | 3.3 | 249.6 | 73.5 | 385.5 |
| | (88.9%) 71 623.6 | (0.5%) 698.2 | (9.3%) 5 349.5 | (1.4%) 2 038.8 | (100%) 79 710.0 | (53.6%) 65 438.0 | (1.1%) 1 296.0 | (40.1%) 40 984.2 | (5.2%) 13 455.8 | (100%) 121 174.0 | (15.3%) 9 901.9 | (0.9%) 428.5 | (64.8%) 20 422.2 | (19.1%) 21 223.0 | (100%) 51 975.7 |

[a] Unit costs are showed on the first row of each activity while

[b] total costs are shown on the second row

[c] Equipment and

[d] supplies lists are detailed in supporting information S2 and S3 Tables

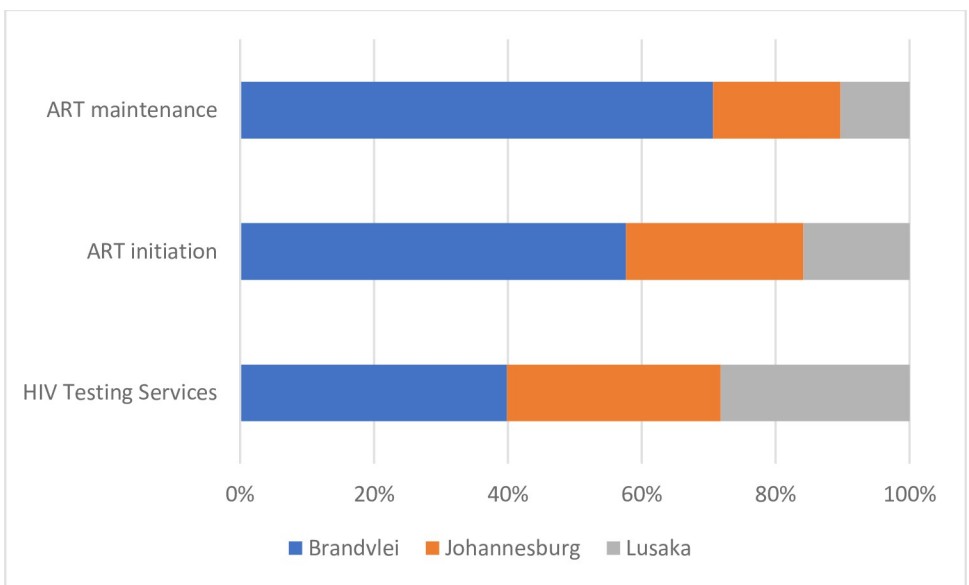

**Fig 1. Summary of overall cost per patient for HIV testing services, ART initiation and ART maintenance.** The largest cost driver for HIV testing at Brandvlei were staff costs at 55.6% of the total cost, while at Johannesburg and Lusaka, supplies were the largest contributor at 56.5% and 86.6% of the total cost, respectively, due to the greater numbers of clients served (S1 Table). The process of conducting the two HIV tests per inmate.

complexities of transferring incarcerated persons from cell blocks to the clinic setting. Correctional wardens and other corrections cadres were involved in HTS to provide security while HIV testing or counselling was delivered by nurses and counsellors. Second, prior HTS costing has generally focused on the clinic setting, where personnel providing HTS may be multi-tasking leading to only the time spent on an HTS session contributing to the HTS cost. This contrasts with HTS screening campaigns or HTS among newly arriving corrections inmates. In the correctional facility setting, HTS personnel may need to wait for additional inmates to be brought to queue or spend set-up time and wait time to provide HTS to a small number of newly incarcerated individuals. Similar decreased efficiency has been reported from community-based and other non-facility HTS settings [14]. The advantage of HTS in the correctional setting is reaching a greater number of people living with HIV, and particularly men and key populations disproportionately affected by HIV/AIDS, many of whom may not visit a health facility when not incarcerated [15].

The cost of ART was also higher in the correctional facility-setting for Brandvlei and Lusaka than a prior report from community clinics in those countries. The annual cost of ART care (not specifying initiation or maintenance) was estimated to be $682 in South Africa and $278 in Zambia [8]. This contrasts to $2,640.6 for Brandvlei, $710 for Johannesburg, and $385.5 for Lusaka. The higher cost at Brandvlei is related to a small volume of ART clients and higher cadres of staff involved in the various components of HIV testing and ART delivery (nurses rather than counsellors). Security requirements were an additional cost as corrections wardens were used to escort clients to and from the clinic and to monitor incarcerated persons while at the clinic. In contrast, standard clinics in the community have minimal or no security and security personnel do not accompany clients to and from places of residence. The cost per client could plausibly decline as the HTS and treatment program matures and ways to achieve greater efficiency are identified such as task shifting to peer educators. However, multiple elements increase costs in the correctional facility, including limited available hours of contact

with incarcerated individuals due to a schedule of when cellblocks are locked down, the need for security escorts, and the staffing numbers based on overall correctional services policies for each facility rather than the workload.

This study had several limitations. Many of the corrections staff involved in UTT have other responsibilities outside of HIV service delivery making it difficult to estimate time spent on HIV service delivery alone. Furthermore, estimation of time spent on UTT activities depended on self-report. Some of the material resources required for UTT were shared with other health services, so we relied on self-report by staff members to assign the proportion of the resource that went to UTT. Unit cost (cost per client) depended on an accurate denominator of the number of clients who received the service for those inputs that were costed top-down. Lastly, the findings need to be considered in context of the correctional facility size, number of people living with HIV, and staffing level of health personnel in the facility. We believe that when considering these factors, reasonable assumptions can be made to apply these findings to other facilities in South Africa or Zambia. However, the recruitment of participants from high and low volume facilities with varied human resource capabilities and HIV prevalence was likely generally representative of correctional facilities in southern Africa.

## Conclusion

As a rule of thumb, reaching the last 20% of individuals for a service is estimated to cost as much as reaching the first 80% [16, 17]. Efforts to reach those who are unreached with HIV testing and ART should include correctional facilities and incarcerated people, despite higher costs. Given the needs of the incarcerated population, the opportunity to reach men at high risk for HIV, and overall national and global 95-95-95 goals, we believe that the cost of providing scaled-up universal test and treatment is worth the potential returns for the health of incarcerated people and the broader community.

## Supporting information

**S1 Table. Service statistics over a 12-month period at correctional facilities in 2017.**
(DOCX)

**S2 Table. Equipment lists for Brandvlei, Johannesburg and Lusaka correctional facilities.**
(DOCX)

**S3 Table. Supplies lists for Brandvlei, Johannesburg and Lusaka correctional facilities.**
(DOCX)

**S1 File. Datasets for Brandvlei, Johannesburg and Lusaka correctional facilities.**
(ZIP)

## Acknowledgments

We would like to thank the teams from South Africa and Zambia for their assistance with data collection and entry namely: Tiro Dinake, Resignation Pelusa, Abraham Jacobus, Phunyezwa Langa, Conzilia Copido, Danielle Daniels-Felix, Vernon van Heerden, Mziwabantu Fosi, Angelina Solomons, Maurihull George, Besa Chibwe, Precious Sakanya and Phillip Chilembo.

## Author Contributions

**Conceptualization:** Rachel Mukora, Helene J. Smith, Michael E. Herce, Harry Hausler, Katherine L. Fielding, Salome Charalambous, Christopher J. Hoffmann.

**Data curation:** Rachel Mukora.

**Formal analysis:** Rachel Mukora, Christopher J. Hoffmann.

**Funding acquisition:** Salome Charalambous, Christopher J. Hoffmann.

**Investigation:** Rachel Mukora, Michael E. Herce, Harry Hausler, Katherine L. Fielding, Salome Charalambous, Christopher J. Hoffmann.

**Methodology:** Rachel Mukora, Helene J. Smith, Michael E. Herce, Lucy Chimoyi, Salome Charalambous, Christopher J. Hoffmann.

**Project administration:** Rachel Mukora, Helene J. Smith, Lucy Chimoyi, Salome Charalambous, Christopher J. Hoffmann.

**Resources:** Helene J. Smith, Michael E. Herce, Lucy Chimoyi, Harry Hausler, Salome Charalambous, Christopher J. Hoffmann.

**Software:** Christopher J. Hoffmann.

**Supervision:** Rachel Mukora, Helene J. Smith, Lucy Chimoyi, Christopher J. Hoffmann.

**Validation:** Rachel Mukora, Helene J. Smith, Michael E. Herce, Lucy Chimoyi, Salome Charalambous, Christopher J. Hoffmann.

**Visualization:** Rachel Mukora, Michael E. Herce, Salome Charalambous, Christopher J. Hoffmann.

**Writing – original draft:** Rachel Mukora, Salome Charalambous, Christopher J. Hoffmann.

**Writing – review & editing:** Rachel Mukora, Helene J. Smith, Michael E. Herce, Lucy Chimoyi, Harry Hausler, Katherine L. Fielding, Salome Charalambous, Christopher J. Hoffmann.

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
