## [Decision Letter · Decision Letter 0]

11 Apr 2022

PONE-D-21-36716Costs of Implementing Universal Test and Treat in Three Correctional Facilities in South Africa and ZambiaPLOS ONE

Dear Dr. Mukora,

Thank you for submitting your manuscript to PLOS ONE. After careful consideration, we feel that it has merit but does not fully meet PLOS ONE’s publication criteria as it currently stands. Therefore, we invite you to submit a revised version of the manuscript that addresses the points raised during the review process.

This paper shows potential given the limited data on health costings in HIV treatment. The reviewers had some concerns however regarding the failure to acknowledge that universal test and treat is a mandate currently. The second reviewer has also recommended that the authors adhere to the CHEERS checklist for health economics studies and also that the tables are clarified.

We look forward to receiving your revised manuscript.

Kind regards,

Elizabeth S. Mayne, M.D.

Academic Editor

PLOS ONE

Journal Requirements:

Additional Editor Comments:

This paper shows potential given the limited data on health costings in HIV treatment. The reviewers had some concerns however regarding the failure to acknowledge that universal test and treat is a mandate currently. The second reviewer has also recommended that the authors adhere to the CHEERS checklist for health economics studies and also that the tables are clarified.

Reviewers' comments:

Reviewer's Responses to Questions

**Comments to the Author**

1. Is the manuscript technically sound, and do the data support the conclusions?

Reviewer #1: Yes

Reviewer #2: Yes

2. Has the statistical analysis been performed appropriately and rigorously? 

Reviewer #1: Yes

Reviewer #2: Yes

3. Have the authors made all data underlying the findings in their manuscript fully available?

Reviewer #1: No

Reviewer #2: Yes

4. Is the manuscript presented in an intelligible fashion and written in standard English?

Reviewer #1: Yes

Reviewer #2: Yes

5. Review Comments to the Author

Reviewer #1: The authors report results from a study that explored the costs of implementing UTT in three correctional facilities, 2 in South Africa and 1 in Zambia. This is an interesting paper given the dearth of costing information particularly related to HIV health-care provision in correctional facilities. The paper is well written and provides a useful perspective although I feel there are some areas that need further expansion. In addition, UTT is now widely rolled out across South Africa and the paper would benefit from further information on how these data will help given that UTT is now standard of care in most health-facilities and whether this is also the case in Zambia and across correctional facilities. I would recommend that references to some of the more recent work on UTT implementation be reviewed and included as I am certain that more has been published in the last five years.

Comments and suggestions:

Line 78 – Introduction – it should be made clear that UTT has now been adopted in both South Africa and Zambia. In fact UTT was introduced in South Africa in September 2016 and recommended same-day ART initiation from October 2017.

• SA-NDOH circular – Implementation of Universal test and treat strategy for HIV positive patients and differentiated care for stable patients. South Africa National Department of Health 2016

• SA NDoH Fast tracking implementation of the 90-90-90 strategy for HIV through implementation of the test and treat (TT) poicy and the sam-day ART initiation for positive patients, 2017

Lines 80-82 – can the authors include stats or what ART uptake, viral load suppression and retention were prior to UTT?

Lines 93-95 – can the authors clarify that these data are from that same reference number 6 and that in this context they are referring to mathematical models that are used not models of care.

Line 104 and paragraph – Methods – one piece of information that would be helpful to include here is a brief description of how inmates at each level of the HIV cascade engage with HIV self-services

Line 127 – Table 1 – it would be helpful if this table could be expanded a bit further to include additional information about each faciity e.g. to incorporate the number of correctional units in those facilities and how these break down into male and female units, maximum security and youth. It might also be helpful to know some of the HIV data for each facility as well if it is known. I had missed the total inmates and staff: inmate ratio rows and wonder if those should not be taken to the top or highlighted in some way

I am not sure if I missed it but it would be helpful if there was supplementary material that also present the various staff costs which might give more context to the differences between facilities and countries. This would provide important context in the results as currently it is difficult to understand Table 2, 3 and 4 where despite the units being quite similar the costs are radically different.

Line 163 – it is not clear what the “(i.e., Correctional services”) is referring to in this line?

Line 170 – I think some punctuation is missing between ART prescription and ART maintenance.

Line 179 – unit costs were calculated as cost per client served. Where did information about the number of clients served come from and could that be incorporate in the tables?

Line 180 – please just clarify that the currencies are ZAR – South African Rand and Zambian Kwacha.

Line 190 – ‘the time spent to complete the activity’ please clarify how this was determined and recorded. Was it recorded for each client i.e. observed or was it estimated and self-reported? This will then also give better context for line 211.

Line 197-198 – where was the number of patients seen over a 12-month period determined from and what was the specific period this was collected for.

Line 207 – please give a reference for the interest rates that were used for South Africa and Zambia.

Line 244 – the restriction to approved study languages in South Africa to English and Afrikaans seems like a limitation. Whilst these languages might have been appropriate for the Western Cape I would expect that for Johannesburg Zulu and Sotho should have been included. Were any staff excluded because they did not speak one of the study languages?

Line 249 – Results how was the number of each cadre of staff at each site determined and how were they selected. This is particularly important to understand how the smallest facility had the largest number of interviews.

Line 263 – I think by including more information in table 1 this might help understand how Brandvlei came to have significantly more staff support HTS than Johannesburg and Lusaka

Line 265 – this point again speaks to there needing to be more information in the text and tables, one assumes that the ART costs should be the same in terms of cost per patient. It is therefore difficult to understand why Jo’burg and Lusaka had much higher supply costs unless they had greater numbers.

Tables 3, 4 and 5 – can the authors just clarify that the first row is unit and the second row is costs. It might also be helpful to include footnotes as to what exactly was included under equipment and supplies.

Was there any thought of disaggregating these costs by gender (or age) e.g. higher or lowers costs for treating male prisoners vs. female, younger vs. older, maximum security vs. general population prisoners?

Line 311 – Discussion - the authors conclude that these data can guide resource allocation for implementing ITT in similar correctional settings. Given that UTT is now implemented as standard of care is this still useful? Is there evidence of slower uptake/implementation in correctional facilities vs primary health care facilities? Would we need more information about these correctional facilities in order to generalize these to other facilities?

Line 344 – is there any new or current evidence that the cost per client did or did not decline 5 years later in the UTT programme?

Line 358 – this relates to a point that I made earlier – if these findings cannot be generalized to all correctional facilities how can they then be used to guide future implementation? If we had more information about these facilities maybe that might help determine where these findings could be used?

Line 365 – Conclusion – please add a reference for your first statement.

Line 370 – again it would be good to know what the level of UTT implementation now is in correctional facilities in SA and Zambia.

Reviewer #2: This is valuable study describing costs of delivering HIV services in a correctional services setting.

Major issues:

Please indicate whether this manuscript complies with the CHEERS checklist for health economic studies (https://bmchealthservres.biomedcentral.com/articles/10.1186/s12913-021-07460-7). In the methods section, some aspects are not adequately described such as outcome of interest, discount rate, etc. From reading the submission, the reference to unit costs appears to be the cost per patient for HST, ART initiation and maintenance. Therefore, it would be more accurate to refer to the total cost and cost per patient across the manuscript. Unit costs refers to cost of a Creatinine medical test for example.

The survey was conducted for the September and December 2017 period. However, the time horizon for the costing analysis is not clearly defined in methods: Is data reported for the January to December 2017 period?

Tables 2 to 4 are difficult to read and could be amended to provide more details. It would easier to read if the total costs, number of patients seen and then the cost per patient are reported. Perhaps the percentage contribution should only be reported for the cost per patient. This would make the costing analysis much easier to follow.

There is a need for a summary figure report the overall cost per patient for HST, ART initiation and ART maintenance for the three correctional facilities.

For the costing data reported for Table 4, please indicate why staffing costs for educational messages was so high at Brandvlei. This is almost 40-fold more expensive than the other correctional facilities. Is this not a calculation error?

Minor issues:

Abstract introduction: HIV is repeated thrice in one sentence – please rephrase (Lines 38-40)

‘Universal test and treat (UTT) for HIV is a population-based strategy that aims to

ensure widespread HIV testing and rapid antiretroviral therapy (ART) for all who have

tested HIV-positive regardless of CD4 count to decrease HIV incidence and improve

health outcomes.’

Abstract conclusion: Abbreviate ‘universal test and treatment’ to UTT (Line 70)

Line 170. Please add a full stop after ‘initial ART prescription’

Line 207: Please provide more details on how equipment costs were annualized.

Table 4: Staff heading missing for the Johannesburg correctional facility.

6. PLOS authors have the option to publish the peer review history of their article (what does this mean?). If published, this will include your full peer review and any attached files.

Reviewer #1: No

Reviewer #2: No

---

## [Author Response · Author response to Decision Letter 0]

25 May 2022

May 23, 2022

Elizabeth S. Mayne

Academic Editor

PLOS ONE

RE: Revision and resubmission of article titled, “Costs of Implementing Universal Test and Treat in Three Correctional Facilities in South Africa and Zambia”

Dear Editor

I would like to thank you for the careful review and invitation to submit a revised version of the manuscript that addresses the points raised during the review process. Please see here the responses to reviewer comments where the lines in the comments below refer to the “Revised Manuscript with Track Changes (all markup)” version of the manuscript. We have also submitted the following documents: 

• 'Revised Manuscript with Track Changes'

• An unmarked version of your revised paper without tracked changes labelled 'Manuscript'

• Supporting information labelled ‘S1_Table.docx’ and ‘S2 and S3_Table.docx’

Comments to the Author

3. Have the authors made all data underlying the findings in their manuscript fully available?

Reviewer #1: No

We have extended Table 1 to include the additional information namely number of correctional units, break down into male and female units, maximum security and youth. We have also included some of the HIV data for each facility namely the estimated HIV prevalence and clients initiated on ART. 

We have included information with the service statistics on the clients served as supporting information. Please see the document labelled ‘S1 _Table.docx’. We have indicated this on line 219.

Reviewer #2: Yes

5. Review Comments to the Author

Reviewer #1: 

Comments and suggestions:

1) Line 78 – Introduction – it should be made clear that UTT has now been adopted in both South Africa and Zambia. In fact UTT was introduced in South Africa in September 2016 and recommended same-day ART initiation from October 2017.

Thank you for the suggestion. We have now added a sentence on lines 80-82 that makes this clear. 

“UTT has been tested in several settings and adopted as part of national HIV programmes in many countries, including both South Africa and Zambia. In fact, UTT was introduced in South Africa in September 2016 and recommended same-day ART initiation from October 2017 (1,2).”

We have also added the suggested articles below to our references:

• SA-NDOH circular – Implementation of Universal test and treat strategy for HIV positive patients and differentiated care for stable patients. South Africa National Department of Health 2016

• SA NDoH Fast tracking implementation of the 90-90-90 strategy for HIV through implementation of the test and treat (TT) policy and the same-day ART initiation for positive patients, 2017

2) Lines 80-82 – can the authors include stats or what ART uptake, viral load suppression and retention were prior to UTT?

We have now included stats on lines 82-87. “Prior to UTT, one retrospective study done using programmatic data in South Africa described viral load suppression at six and 12 months after ART initiation at 94.7% and 92.5% respectively for incarcerated people with HIV treated in an on-site ART programme(3). A cross sectional study done in Malawi also prior to UTT, found 95% viral suppression among a sample of incarcerated people with HIV who received ART for at least 6 months in a prison clinic(4).” 

There is a lack of data on overall ART coverage in correctional facilities prior to UTT.

3) Lines 93-95 – can the authors clarify that these data are from that same reference number 6 and that in this context they are referring to mathematical models that are used not models of care.

We apologise for this oversight. We have provided the reference number 10 (originally number 6) on line 105 and we have also clarified that we are referring to mathematical models. 

4) Line 104 and paragraph – Methods – one piece of information that would be helpful to include here is a brief description of how inmates at each level of the HIV cascade engage with HIV self-services

On lines 114-120, we have included a brief description on how inmates at each level of the HIV cascade engaged with HIV self-services.

5) Line 127 – Table 1 – it would be helpful if this table could be expanded a bit further to include additional information about each facility e.g. to incorporate the number of correctional units in those facilities and how these break down into male and female units, maximum security and youth. It might also be helpful to know some of the HIV data for each facility as well if it is known. I had missed the total inmates and staff: inmate ratio rows and wonder if those should not be taken to the top or highlighted in some way.

We have extended Table 1 to include the additional information namely number of correctional units, break down into male and female units, maximum security and youth. We have also included some of the HIV data for each facility namely the estimated HIV prevalence and clients initiated on ART. 

6) I am not sure if I missed it but it would be helpful if there was supplementary material that also present the various staff costs which might give more context to the differences between facilities and countries. This would provide important context in the results as currently it is difficult to understand Table 2, 3 and 4 where despite the units being quite similar the costs are radically different.

We have included information with the service statistics on clients served as supporting information. Please see the document labelled ‘S1 _Table.docx’. We have indicated this on line 219.

7) Line 163 – it is not clear what the “(i.e., Correctional services”) is referring to in this line?

We have deleted “(i.e., Correctional services”) on lines 181-182 to avoid any confusion.

8) Line 170 – I think some punctuation is missing between ART prescription and ART maintenance.

We have included a full stop between ART prescription and ART maintenance on line 189.

9) Line 179 – unit costs were calculated as cost per client served. Where did information about the number of clients served come from and could that be incorporated in the tables?

The information on the clients served came from the TasP study. We have included information with the service statistics on clients served as supporting information. Please see the document labelled ‘S1 _Table.docx’. We have indicated this on line 219.

10) Line 180 – please just clarify that the currencies are ZAR – South African Rand and Zambian Kwacha.

We have provided clarity on lines 198-200.

“All costs were converted into the US dollar (USD) using exchange rates on 1 January 2017: 1USD = 13.78 South African Rand (ZAR) for the South African sites and 1USD = 9.90 Zambian Kwacha (ZMW) for the Zambian site.”

11) Line 190 – ‘the time spent to complete the activity’ please clarify how this was determined and recorded. Was it recorded for each client i.e. observed or was it estimated and self-reported? This will then also give better context for line 211.

The time spent per client to complete the activity was estimated and self-reported. I have specified this on line 210.

12) Line 197-198 – where was the number of patients seen over a 12-month period determined from and what was the specific period this was collected for.

The number of clients seen over a 12-month period was determined from the service statistics obtained from the TasP study. I have included this on line 219. They were collected over a 17-month period (September 2016 – March 2018) so we worked out the average number of clients seen over a 12-month period.

13) Line 207 – please give a reference for the interest rates that were used for South Africa and Zambia.

I have added a reference for the interest rate on lines 230 -231 and edited the text for clarity.

14) Line 244 – the restriction to approved study languages in South Africa to English and Afrikaans seems like a limitation. Whilst these languages might have been appropriate for the Western Cape I would expect that for Johannesburg Zulu and Sotho should have been included. Were any staff excluded because they did not speak one of the study languages?

Since the study on the costs was only conducted to staff, we did not believe that this would lead to exclusion of any staff members. In fact, no staff in Johannesburg or Brandvlei were excluded from the study due to language barriers as all of them spoke English. 

15) Line 249 – Results how was the number of each cadre of staff at each site determined and how were they selected. This is particularly important to understand how the smallest facility had the largest number of interviews.

On lines 154-156 we have included the following methods that were used to determine and select the number of each cadre of staff at each site.

“We purposively selected all staff within the cadre where there were two or less staff in the role. In cadres where three or more staff performed the same duties, we used convenience sampling where every 2nd person was selected.”

16) Line 263 – I think by including more information in table 1 this might help understand how Brandvlei came to have significantly more staff support HTS than Johannesburg and Lusaka.

We apologise for any lack of clarity but we would like to clarify that at the bottom of table 1 we included 42 staff at Brandvlei with a staff to inmate ratio of 1:71, Johannesburg had 62 staff and a staff to inmate ratio of 1:65 while Lusaka had 78 staff and a staff to inmate ratio of 1:18. Based on these numbers, it appears that Brandvlei did not have more staff support HTS but instead Lusaka did.

We have specified the rationale for the staffing on lines 140-141. “The rationale for the staffing in SA was based on the national policy regarding staff per facility rather than on need.”

17) Line 265 – this point again speaks to there needing to be more information in the text and tables, one assumes that the ART costs should be the same in terms of cost per patient. It is therefore difficult to understand why Jo’burg and Lusaka had much higher supply costs unless they had greater numbers.

Yes, the total costs of supplies for HTS are higher at Jo’burg and Lusaka due to the greater numbers of clients served. I have included information with the service statistics on the number of clients served as supporting information. Please see the document labelled ‘S1 _Table.docx’. 

On lines 291-294, I have also included additional text in the manuscript to clarify this. “The largest cost driver for HIV testing at Brandvlei were staff costs at 55.6% of the total cost, while at Johannesburg and Lusaka, supplies were the largest contributor at 56.5% and 86.6% of the total cost, respectively, due to the greater numbers of clients served (supplementary information ‘S1 _Table.docx’).”

18) Tables 3, 4 and 5 – can the authors just clarify that the first row is unit and the second row is costs. It might also be helpful to include footnotes as to what exactly was included under equipment and supplies.

We have clarified that the first row is aunit costs and the second row is btotal costs using footnotes.

We have included detailed lists of cequipment and dsupplies in supporting information and we have indicated this within the footnotes. Please see the document labelled ‘S2 and S3 _Table.docx’.

19) Was there any thought of disaggregating these costs by gender (or age) e.g. higher or lower costs for treating male prisoners vs. female, younger vs. older, maximum security vs. general population prisoners?

Thank you for this question. Unfortunately, we did not anticipate that the costs would vary by gender, age or maximum vs general population prisoners so our survey questions did not specify these characteristics.

20) Line 311 – Discussion - the authors conclude that these data can guide resource allocation for implementing UTT in similar correctional settings. Given that UTT is now implemented as standard of care is this still useful? Is there evidence of slower uptake/implementation in correctional facilities vs primary health care facilities? Would we need more information about these correctional facilities in order to generalize these to other facilities?

Given that UTT is now policy this evidence is still useful for resource allocation for implementation. Adding responsibilities without appropriate resources can lead to limited implementation of policy. Our study also contributes to early evidence that can be used as a comparator for future studies.

To our knowledge, there are no studies comparing UTT implementation in correctional facilities vs primary health care facilities. It is plausible that implementation is less complete in some correctional facilities due to resources and staff priorities.

We do not need any additional information about these correctional facilities in order to generalize these findings to other facilities.

21) Line 344 – is there any new or current evidence that the cost per client did or did not decline 5 years later in the UTT programme?

To our knowledge there is no current evidence that the cost per client did or did not decline 5 years later in the UTT programme.

Our results do provide a point of comparison for future studies.

22) Line 358 – this relates to a point that I made earlier – if these findings cannot be generalized to all correctional facilities how can they then be used to guide future implementation? If we had more information about these facilities maybe that might help determine where these findings could be used?

On lines 389-393, we have revised this statement: “The findings need to be considered in context of the correctional facility size, number of people living with HIV, and staffing level of health personnel in the facility.” We believe that when considering these factors, reasonable assumptions can be made to apply these findings to other facilities in South Africa or Zambia.

23) Line 365 – Conclusion – please add a reference for your first statement.

We have added the following references to the statement on line 399-400.

Hill LG, Maucione K, Hood BK. A focused approach to assessing program fidelity. Prev Sci. 2007 Mar;8(1):25-34. doi: 10.1007/s11121-006-0051-4. Epub 2006 Sep 13. PMID: 16967341.

Harolds, Jay MD Quality and Safety in Health Care, Part I, Clinical Nuclear Medicine: August 2015 - Volume 40 - Issue 8 - p 660-662 doi: 10.1097/RLU.0000000000000877

24) Line 370 – again it would be good to know what the level of UTT implementation now is in correctional facilities in SA and Zambia.

We agree that a follow-up study or program evaluation would be valuable. We don’t have any recent assessments to include or report for this manuscript.

Reviewer #2: 

Major issues:

1) Please indicate whether this manuscript complies with the CHEERS checklist for health economic studies (https://bmchealthservres.biomedcentral.com/articles/10.1186/s12913-021-07460-7). In the methods section, some aspects are not adequately described such as outcome of interest, discount rate, etc. From reading the submission, the reference to unit costs appears to be the cost per patient for HST, ART initiation and maintenance. Therefore, it would be more accurate to refer to the total cost and cost per patient across the manuscript. Unit costs refers to cost of a Creatinine medical test for example.

We apologise if some aspects of the CHEERS checklist may not have been adequately described in our manuscript. We have now clarified our outcomes of interest (lines 180 - 181). The discount rate was initially included in the original version and has now been edited slightly for clarity (lines 230 - 231).

Thank you for the second suggestion. We would like to clarify that we defined the ‘units’ in the unit costs as the ‘clients served’ which we mentioned on line 202. “Costing results are reported separately for each facility and as overall cost for care provision and as cost per client served (unit cost).”

To guide us, we used the following definition from the Reference case written by the Global Health Costing Consortium (GHCC).

Intervention ‘unit’ cost Average cost of an intervention (or strategy) (e.g., unit cost per person or episode of expanding TB treatment, or costs of peer education per person reached)

2) The survey was conducted for the September and December 2017 period. However, the time horizon for the costing analysis is not clearly defined in methods: Is data reported for the January to December 2017 period?

The costing analysis is reported over a 12-month period with no time horizon specified because the number of clients seen were collected over a 17-month period (September 2016 – March 2018) so we worked out our costs based on the average number of clients seen over a 12-month period.

We have included the following statement on lines 218 – 221 to make this clear.

“Staff unit costs were computed by dividing the annual total costs with the number of clients seen over a 12-month period which we obtained from the TasP study (S1_Table.docx) for a period of 17 months, though we based our costs on the average number of clients seen over a 12-month period.”

3) Tables 2 to 4 are difficult to read and could be amended to provide more details. It would easier to read if the total costs, number of patients seen and then the cost per patient are reported. Perhaps the percentage contribution should only be reported for the cost per patient. This would make the costing analysis much easier to follow.

On tables 2 to 4 we have now reported the total costs and cost per patient. The clients seen are reported as supporting information. Please see the document labelled ‘S1 _Table.docx’. The percentage contribution is only reported for cost per patient.

4) There is a need for a summary figure report the overall cost per patient for HST, ART initiation and ART maintenance for the three correctional facilities.

We have included a summary figure 1 reporting the overall cost per patient for HST, ART initiation and ART maintenance in Fig 1 (lines 287 - 289).

5) For the costing data reported for Table 4, please indicate why staffing costs for educational messages was so high at Brandvlei. This is almost 40-fold more expensive than the other correctional facilities. Is this not a calculation error?

The high staffing costs for educational messages at Brandvlei was due to one of our study limitations because of using self-reported time data. We have mentioned this on line 384-385. ‘Furthermore, estimation of time spent on UTT activities depended on self-report’.

Minor issues:

6) Abstract introduction: HIV is repeated thrice in one sentence – please rephrase (Lines 38-40)

‘Universal test and treat (UTT) for HIV is a population-based strategy that aims to

ensure widespread HIV testing and rapid antiretroviral therapy (ART) for all who have

tested HIV-positive regardless of CD4 count to decrease HIV incidence and improve

health outcomes.’

We have edited the sentence on lines 38-40 and reduced the repetition of the word HIV.

7) Abstract conclusion: Abbreviate ‘universal test and treatment’ to UTT (Line 70)

We have abbreviated ‘universal test and treatment’ to UTT (Line 69). We have also revised the statement on lines 68 – 71 to the following:

“Given the needs of this population, the opportunity to reach inmates at high risk for HIV, and overall national and global 95-95-95 goals, the UTT policies for incarcerated individuals are of vital importance. Our findings provide comparator costing data and highlight key drivers of UTT cost by facility.”

8) Line 170. Please add a full stop after ‘initial ART prescription’

We have added a full stop after ‘initial ART prescription’ on line 189.

9) Line 207: Please provide more details on how equipment costs were annualized.

On line 230-231 we have provided a reference for the discount rate used to annualize the equipment costs.

10) Table 4: Staff heading missing for the Johannesburg correctional facility.

We have now included it.

Additional revisions

1) Formatted spacing on line 130

2) Edited footnote beneath table 1 on line 147 for clarity

3) Formatted manuscript to PLOS ONE requirements namely heading levels 1,2 and 3

4) Edited the title page according to PLOS ONE's style requirements

5) We have edited the term “patients” to “clients” throughout the manuscript

---

## [Decision Letter · Decision Letter 1]

22 Jul 2022

Costs of implementing Universal Test and Treat in three correctional facilities in South Africa and Zambia

PONE-D-21-36716R1

Dear Dr. Mukora,

We’re pleased to inform you that your manuscript has been judged scientifically suitable for publication and will be formally accepted for publication once it meets all outstanding technical requirements.

Kind regards,

Elizabeth S. Mayne, M.D.

Academic Editor

PLOS ONE

Additional Editor Comments (optional):

Reviewers' comments:

Reviewer's Responses to Questions

**Comments to the Author**

1. If the authors have adequately addressed your comments raised in a previous round of review and you feel that this manuscript is now acceptable for publication, you may indicate that here to bypass the “Comments to the Author” section, enter your conflict of interest statement in the “Confidential to Editor” section, and submit your "Accept" recommendation.

Reviewer #1: All comments have been addressed

Reviewer #2: All comments have been addressed

2. Is the manuscript technically sound, and do the data support the conclusions?

Reviewer #1: Yes

Reviewer #2: Yes

3. Has the statistical analysis been performed appropriately and rigorously? 

Reviewer #1: Yes

Reviewer #2: Yes

4. Have the authors made all data underlying the findings in their manuscript fully available?

Reviewer #1: Yes

Reviewer #2: Yes

5. Is the manuscript presented in an intelligible fashion and written in standard English?

Reviewer #1: Yes

Reviewer #2: Yes

6. Review Comments to the Author

Reviewer #1: The authors have done well address the reviewer comments that were made and have added clarity where needed. A couple of minor points that might be addressed.

Table 1 might need an updated title given the inclusion of the additional data. Perhaps also to check the HIV prevalence for the Lusaka correctional facility as it is given as 15% yet 16% of the inmates are currently on ART suggesting that HIV prevalence might be higher. Likewise it might be worth mention that in the other correctional facilities based on the proportion that are on ART (2% in Brandvlei and 6% in Johannesburg) it suggests that there are large numbers that are not on ART and this must be in part what is driving up the cost as there are a large number of people serving a very few people on ART. I think it is worthy of discussion that not only is the variation in costs the result of differences in the staff:inmate ratio and variation of salary but must also be driven by the variation in the inmate population (Brandvlei looks to have a higher proportion of female inmates and no maximum security units compared to the other two) which in turn might explain the difference in HIV prevalence and low yield of HIV positive.

The only other point I would make - where the authors discuss supplies as a cost driver they link this to 'the number of clients served' - this to me indicates the number of client who receive care or services whereas I think that often the authors are specifically meaning the number of clients tested or the number supported on ART. I think it is worthwhile being specific about these and what is meant in each case.

Reviewer #2: All comments that were raised have been addressed and there no further changes required for this manuscript.

7. PLOS authors have the option to publish the peer review history of their article (what does this mean?). If published, this will include your full peer review and any attached files.

Reviewer #1: No

Reviewer #2: **Yes: **Naseem Cassim

---

## [Editor Report · Acceptance letter]

16 Aug 2022

PONE-D-21-36716R1 

Costs of implementing Universal Test and Treat in three correctional facilities in South Africa and Zambia 

Dear Dr. Mukora:

I'm pleased to inform you that your manuscript has been deemed suitable for publication in PLOS ONE. Congratulations! Your manuscript is now with our production department. 

Kind regards, 

on behalf of

Dr. Elizabeth S. Mayne 

Academic Editor

PLOS ONE